# Evaluation of the Effect of β-Wrapin AS69 in a Mouse Model Based on Alpha-Synuclein Overexpression

**DOI:** 10.3390/biom14070756

**Published:** 2024-06-25

**Authors:** Lennart Höfs, David Geißler-Lösch, Kristof M. Wunderlich, Eva M. Szegö, Chris Van den Haute, Veerle Baekelandt, Wolfgang Hoyer, Björn H. Falkenburger

**Affiliations:** 1Department of Neurology, Technische Universität Dresden, 01307 Dresden, Germanydavid.geissler-loesch@ukdd.de (D.G.-L.);; 2Deutsches Zentrum für Neurodegenerative Erkrankungen (DZNE), 01307 Dresden, Germany; 3Leuven Viral Vector Core, KU Leuven, 3000 Leuven, Belgium; 4Laboratory for Neurobiology and Gene Therapy, Department of Neurosciences, Leuven Brain Institute, KU Leuven, 3000 Leuven, Belgium; 5Institut für Physikalische Biologie, Heinrich-Heine-Universität Düsseldorf, 40225 Düsseldorf, Germany; 6Institute of Biological Information Processing (IBI-7), Forschungszentrum Jülich GmbH, 52428 Jülich, Germany

**Keywords:** α-Synuclein, alpha-Synuclein, Parkinson’s disease, PD, AS69, β-wrapin, aggregation inhibitors, amyloid aggregation inhibitors, recombinant adeno-associated viral vector, rAAV, synuclein overexpression, Substantia nigra, mouse model, in vivo

## Abstract

Aggregation of the protein α-Synuclein (αSyn) is a hallmark of Parkinson’s disease (PD), dementia with Lewy bodies (DLB) and multiple systems atrophy, and alleviating the extent of αSyn pathology is an attractive strategy against neurodegeneration. The engineered binding protein β-wrapin AS69 binds monomeric αSyn. AS69 reduces primary and secondary nucleation as well as fibril elongation in vitro. It also mitigates aSyn pathology in a mouse model based on intrastriatal injection of aSyn pre-formed fibrils (PFFs). Since the PFF-based model does not represent all aspects of PD, we tested here whether AS69 can reduce neurodegeneration resulting from αSyn overexpression. Human A53T-αSyn was overexpressed in the mouse Substantia nigra (SN) by using recombinant adeno-associated viral vector (rAAV). AS69 was also expressed by rAAV transduction. Behavioral tests and immunofluorescence staining were used as outcomes. Transduction with rAAV-αSyn resulted in αSyn pathology as reported by phospho-αSyn staining and caused degeneration of dopaminergic neurons in the SN. The co-expression of rAAV-AS69 did not reduce αSyn pathology or the degeneration of dopaminergic neurons. We conclude that αSyn monomer binding by rAAV-AS69 was insufficient to protect from aSyn pathology resulting from αSyn overexpression.

## 1. Introduction

Parkinson’s disease (PD), multiple systems atrophy and dementia with Lewy bodies (LBD) are characterized by the aggregation of the small synaptic protein α-Synuclein (αSyn). The αSyn proteinopathy spreads between neurons and glial cells as well as interconnected brain regions. Alleviating the extent of αSyn pathology, e.g., by preventing the formation of synuclein aggregates, is therefore considered an attractive strategy against neurodegeneration in PD and related disorders [1,2].

AS69 is an engineered dimeric protein that induces a local folding of αSyn residues 37–54 into a β-hairpin [3]. It is thereby able to prevent primary and secondary nucleation as well as fibril elongation by sequestering monomeric αSyn [3,4]. AS69′s ability to inhibit the formation of αSyn aggregates has been demonstrated in HEK293T-cells and a *Drosophila melanogaster* model based on neuronal αSyn overexpression [4]. Furthermore, AS69 alleviates αSyn pathology in primary mouse neurons and in a mouse model based on intrastriatal injection of preformed αSyn fibrils [5]. Small molecules that inhibit the aggregation of αSyn, such as UCB0599 and Anle138b, are being studied in clinical trials for Parkinson’s disease (NCT04685265, NCT04875962).

Injecting αSyn fibrils into mouse brains can induce αSyn pathology and dopaminergic neuron death [6]. Yet, fibril-based models only reproduce a part of αSyn pathology [7,8]. In patients, PD can be triggered by the increased expression of αSyn-resulting from either triplication of the αSyn locus or from genetic variants in the αSyn promoter region [9,10,11]. Overexpression of aSyn therefore constitutes another realistic model of PD pathogenesis. In order to determine the utility of monomer binding, we tested AS69 expressed by transduction with a recombinant adeno-associated vector (rAAV) in a mouse model that is based on rAAV-mediated overexpression of human A53T-αSyn (H-αSyn) in the Substantia nigra (SN).

## 2. Materials and Methods

### 2.1. Animals

All animal experiments were carried out in accordance with the European Communities Council Directive of 24 November 1986 (86/609/EEC) and approved by the Landesdirektion Dresden, Germany (25-5131/49615), on 5 May 2020. Twenty 12-week-old C57BLl/6 male mice from Charles River Laboratories (Wilmington, MA, USA) were housed under a 12 h light and dark cycle with free access to pellet food and water in the Experimental Center, TU Dresden, Dresden, Germany.

### 2.2. Recombinant rAAV Production and Purification

The production of rAAVs was performed at the Leuven Viral Vector Core as described previously [12]. Pseudotype rAAV2/7 was used due to its efficient uptake in neurons. The ssDNA virus encodes for the following sequences: ITR-CMVie-enhanced synapsin1 promoter-transgene-WPRE-bovine growth hormone polyadenylation sequence-ITR. Either green fluorescent protein (GFP), human A53T-αSyn or AS69 were inserted as a transgene. As technical titers, genomic copies (GC) were determined by real-time PCR analysis. All viral vectors were diluted to a concentration of 7 × 10^11^ GC/mL. Viral vector solutions were aliquoted and freshly thawed before stereotactic injection. The GFP control cohort received 2 µL of rAAV-GFP, while the other groups received a 2 µL of a 1:1 mix of the respective viral vectors.

### 2.3. Stereotactic Surgery and Tissue Processing

Stereotactic surgeries and the injection of viral vector solutions were performed unilaterally into the SN. The following stereotactic coordinates relative to Bregma were used: anterior-posterior −3 mm; lateral −1.2 mm; and dorso-ventral −4.1 mm. All procedures were performed using aseptic techniques. Mice were anesthetized using Ketamine (100 mg/kg i.p) and Xylazine (10 mg/kg i.p.). Animals were placed in a flat skull position in a stereotactic head frame (Stoelting Co., Wood Dale, IL, USA). A bore hole craniotomy was performed and was followed by a dural incision. A 30 G needle and a 10 µL Hamilton syringe (Hamilton Bonaduz AG, Bonaduz, Switzerland) were used for the injection of 2 µL vector solution. The needle was advanced at a speed of 0.3 mm/min and was left in place at the final coordinates for 5 min before and after the injection of the vector. The vector dilution was injected at a rate of 200 nl/min. After a slow withdrawal, the skin was sutured and lidocaine (Aspen Pharmacare, Durban, South Africa) was topically applied. Eight weeks later, mice were sacrificed with an overdose of isoflurane (Baxter, Lessines, Belgium) and transcardially perfused with 4% paraformaldehyde (PFA) diluted in Tris-buffered saline (TBS, pH 7.6). The brains were kept in PFA 4% for another 48 h at 4 °C and were afterwards transferred to 30% sucrose in TBS for cryoprotection. The brains were frozen at −55 °C in isopentane and stored at −80 °C until 30 µm thick coronal brain sections were obtained using a Cryostat (Leica Biosystems, Nussloch, Germany).

### 2.4. Cylinder Test

The cylinder test was performed at three time points: one day before stereotactic injection and four weeks and eight weeks after surgery. One animal at a time was transferred into a 15 cm wide glass cylinder and up to 25 contacts with the cylinder wall were videotaped. The time until the dominant forepaw performed 25 contacts was scored based on the videos. 

### 2.5. Immunofluorescence Staining

The brain sections were stained for the following antigens: tyrosine hydroxylase (TH), human αSyn (H-αSyn), phosphorylated αSyn at Serine 129 (P-αSyn), glial fibrillary acidic protein (GFAP) and ionized calcium-binding adapter molecule 1 (Iba1). First, the sections were rinsed three times for 10 min in TBS and afterwards incubated in 10% donkey serum (BIOZOL Diagnostica Vertrieb GmbH, Eching, Germany) and 0.2% Triton X-100 (Thermo Scientific, Waltham, MA, USA) diluted in TBS for 1 h at room temperature. Next, the sections were incubated for 18 h in sheep anti-TH (1:2000, P60101; Pel-Freez, Rogers, AR, USA), rabbit anti-P-αSyn (1:1000, ab51253; Abcam, Cambridge, UK), rat anti-H-αSyn (1:1000, ALX-804-258-L001; ENZO Life Sciences, Farmingdale, NY, USA), chicken anti-GFAP (1:500, ab4674, Abcam) or guinea pig anti-Iba1 (1:2000, HS-234308; Histo Sure, Göttingen, Germany) at 4 °C. After three additional 10 min rinses, fluorophore-conjugated secondary antibodies were added for 1 h at 21 °C: Alexa 647-conjugated donkey anti-sheep (1:1000, A21448; Molecular Probes, Eugene, OR, USA), donkey anti-rat 647 (1:1000, ab150155, Abcam), Alexa 647-conjugated donkey anti-chicken (1:500, 703-605-155; Jackson ImmunoResearch Laboratories, West Grove, PA, USA), Alexa 555-conjugated donkey anti-rabbit (1:1000, A31572, Molecular Probes) and Alexa 555-conjugated donkey anti-guinea pig (1:1000, A-21435, Molecular Probes). Nuclei were counterstained with Hoechst (1:2000 for 5 min; Invitrogen, Waltham, MA, USA) and the sections were mounted in Fluoromount-G (Invitrogen, Waltham, MA, USA).

### 2.6. Quantification of TH-Positive Neurons, Analysis of TH-Positive Dendrites and Striatal TH-Positive Axon Terminals

For the quantification of dopaminergic neurons in the SN, every fourth section (about 10 sections per animal) of the SN was stained for TH. Slide scans were acquired by the Light Microscopy Facility (DZNE, Bonn, Germany) using a slide scanning microscope-AxioScan.Z1 (Zeiss, Jena, Germany) equipped with a 20×/0.8NA objective. Five Z-levels were imaged at intervals of one micrometer. TH-positive neurons in the Substantia nigra pars compacta (SNc) were counted manually and investigator-blinded in each hemisphere using Zeiss Zen 3.1 software (Zeiss, Jena, Germany). To obtain an estimate of the total number of TH-positive neurons per hemisphere, the number of TH-positive SNc neurons of all slices was summed; the sum of neurons in the injected hemispheres was expressed relative to the uninjected hemisphere.

Dopaminergic neurons extend long apical dendrites into the Substantia nigra pars reticulata (SNr). The density of these dendrites was assessed by measuring the average intensity of TH staining within each hemisphere’s SNr in every slide scan mentioned above. The signal intensity was averaged across each hemisphere for each animal. Finally, mean intensity in the injected hemispheres was expressed relative to the uninjected hemisphere.

Dopaminergic neurons project their axons into the striatum. In order to analyze their integrity, we collected ten to fifteen high-magnification images from two to three sections from the dorsal striatum using a spinning disk confocal microscope. The setup uses a Zeiss Axio Observer.Z1 Inverted Microscope (Zeiss, Jena, Germany) and a Yokogawa CSU-X1 unit (Yokogawa Life Science, Musashino-shi, Tokyo, Japan) running a 40×/0.95 objective. Image analysis was performed in CellProfiler using the unbuilt “Enhance Neurites” function followed by thresholding the image using the “robust background” algorithm [13]. Next, we calculated for each animal the relative mean count of positive pixels per hemisphere and presented the values from the injected hemisphere normalized to the contralateral hemisphere.

### 2.7. Evaluation of αSyn Pathology

The expression of H-αSyn and P-αSyn within the SNc was analyzed by the following protocol: Scans from every fourth section were acquired using a spinning disk confocal microscope. The setup uses a Zeiss Axio Observer.Z1 Inverted Microscope (Zeiss) and a Yokogawa CSU-X1 unit (Yokogawa Life Science, Musashino-shi, Tokyo) running a 20×/0.8NA objective. Five Z-levels at intervals of 1 µm were scanned per field of view. Maximum intensity projections were created. In order to quantify the number of H-αSyn- and P-αSyn-positive cells, we worked with an open-source software program, namely QuPath [14]. First, the SNc within the treated hemisphere was carefully encircled. Next, the cell detection function was applied using its default parameters, which detects nuclei based on their Hoechst staining. We manually annotated examples for H-αSyn-positive cells and P-αSyn-positive cells as well as a background signal in every SNc which allowed us to train the in-build object classifier algorithm which analyzed the whole SNc based on our manually annotated examples. The count of positive cells (H-αSyn and P-αSyn) per animal was summed and the sum was multiplied by four since every fourth section was analyzed.

### 2.8. Analysis of Astrogliosis and Microgliosis

For the quantification of astrogliosis and microgliosis, fifteen to twenty images within the SNc of both hemispheres per animal spanning four sections were acquired using the spinning disk confocal microscope described above with a 40×/0.95NA objective. Seven z-levels at a step size of 0.48 µm were covered per image. Images were processed as maximum intensity projections. The GFAP channel was thresholded via a “robust background method” in CellProfiler and the resulting binary images were opened once to reduce salt-and-pepper noise. The count of positive pixels was divided by the total count of pixels per image. Furthermore, the binary image was used to create a masked image of the original GFAP channel and the signal intensity in the GFAP-positive area was determined. We calculated the mean GFAP-positive area and the mean GFAP intensity per hemisphere and per animal. The results of the injected hemisphere were expressed relative to the uninjected hemisphere.

In order to evaluate the degree of microglia activation, we analyzed the Iba1 channel from the same images as described for GFAP (15–20 images per animal from four sections and both hemispheres). The analysis was performed manually by classifying each Iba1-positive cell into one of the three following categories based on its morphology: (I) ramified, (II) hypertrophic or dystrophic or (III) amoeboid or rod-shaped microglia [15]. Next, we determined the mean count of microglia per image for each animal and hemisphere. The average number of microglia per image in the injected hemisphere is shown relative to the average per image number of microglia in the uninjected hemisphere. We also show, based on data from the injected hemisphere, the mean count of each microglia subtype relative to the mean count of all microglia categories. 

### 2.9. Statistical Analysis

Data analysis and visualization were performed in R (version 4.2.0, R Core Team, Vienna, Austria) using RStudio (“Spotted Wakerobin” Release, 7872775e, 22 July 2022, RStudio Team, Boston, MA, USA) and the following packages: Tidyverse for data wrangling and visualization; and Ggpubr, FSA and Corrplot for statistical testing [16,17,18,19,20,21]. The Shapiro–Wilk normality test was used to determine the normal distribution of the data and the Bartlett’s test was used to assess equal population variances. For normally distributed data, we applied Welch’s two-sample *t*-test, Welch’s two-sample *t*-tests with Benjamini–Hochberg *p*-value correction for multiple testing or a two-way ANOVA followed by Tukey’s HSD depending on the number of comparisons. Data that did not meet above-described standards were compared using the Wilcoxon rank sum exact test for pairwise comparisons or the Kruskal–Wallis test with Holm’s *p*-value correction for multiple testing followed by Dunn’s test as post hoc test. In order to assess correlations, we calculated Pearson’s product–moment correlation. Error bars represent mean ± SEM. Exact *p*-values are listed in the figure legends.

## 3. Results

### 3.1. Overexpression of αSyn in Mouse SN Induced a Subtle Motor Deficit

In this study, we induced αSyn pathology and neurodegeneration in mice by transducing SN neurons with human A53T-αSyn using rAAV vectors [12,22]. Transduction with rAAV-GFP was used as the negative control. Cylinder tests were performed at three time points: one day before stereotactic injections and 4 weeks as well as 8 weeks post-injection (Figure 1A). We measured the time it took each animal to touch the cylinder wall 25 times (Figure 1B). At baseline, it took all groups equally long. At 4 weeks, it took animals transduced with rAAV-αSyn + rAAV-GFP (blue in Figure 1B) somewhat longer to complete 25 contacts than observed in the other groups, but the difference was not statistically significant. At eight weeks, mice transduced with rAAV-αSyn + rAAV-GFP required on average almost twice as long to complete 25 contacts than controls did. We conclude that αSyn overexpression induced deficits in animals’ motor abilities.

Furthermore, we studied the consequences of transducing SN neurons with AS69, also using rAAV vectors. Again, transduction with rAAV-GFP was used as the negative control, resulting in four treatment groups (Figure 1A). The cylinder test performance of mice transduced with rAAV-AS69 + rAAV-GFP did not differ from mice transduced with rAAV-GFP only. In summary, mice transduced with rAAV-αSyn + rAAV-GFP resulted in a slower phenotype, which was somewhat restored in mice transduced with rAAV-αSyn + rAAV-AS69 (Figure 1B). Yet, this difference was not statistically significant.

### 3.2. rAAV-AS69 Did Not Prevent the Degeneration of TH-Positive Neurons

Next, we quantified the number of dopaminergic neurons in the SNc, using tyrosine hydroxylase (TH) as a marker (Figure 2A). Hemispheres transduced with rAAV-GFP showed a similar number of TH-positive neurons as uninjected hemispheres (Figure 2B), suggesting that transduction with rAAV-GFP did not cause the degeneration of dopaminergic neurons in the SN. This also holds true for mice transduced with rAAV-GFP + rAAV-AS69. Mice transduced with rAAV-αSyn + rAAV-GFP showed about 20% fewer TH-positive neurons than rAAV-GFP-expressing animals. The reduction in dopaminergic neurons by αSyn overexpression is therefore modest, but it can explain the subtle motor deficit (Figure 1B). In mice transduced with rAAV-αSyn + rAAV-AS69, the loss of TH-positive neurons was similar to that in mice with rAAV-αSyn + rAAV-GFP (Figure 2B). This suggests that transduction with rAAV-AS69 did not reduce the αSyn-induced loss of dopaminergic neurons in the SN.

We also analyzed the immunoreactivity of the long apical dendrites of dopaminergic neurons in the SNr (Figure 2C). Specifically, the mean signal intensity in the TH channel within the SNr was normalized to the contralateral hemisphere. Normalized TH immunoreactivity in the SNr was significantly reduced in mice transduced with rAAV-αSyn as compared to rAAV-GFP but not significantly different between mice transduced with rAAV-αSyn + rAAV-AS69 compared to rAAV-αSyn + rAAV-GFP (Figure 2C), consistent with the observations in the SNc. 

Moreover, we analyzed the density of TH-positive axon terminals in the striatum (Figure 2D,E). The density of TH-positive axon terminals in the striatum was expressed relative to the intact contralateral hemisphere. With this method, we did not observe the degeneration of axon terminals with αSyn overexpression (Figure 2E).

### 3.3. Transduction with rAAV-AS69 Is Associated with an Increase in αSyn Pathology

In order to better understand the consequences of transducing SN neurons with human αSyn (H-αSyn), we stained for H-αSyn and quantified the extent of αSyn pathology using p-S129-αSyn (P-αSyn) as an established epitope for pathologically aggregated αSyn [23,24,25]. We acquired slide scans of the SN (Figure 3A) for the quantification of positive cells (Figure 3C) as well as high-magnification images of dopaminergic neurons co-stained for H-αSyn and P-αSyn (Figure 3B). The count of H-αSyn-positive cells seemed to be increased in animals transduced with rAAV-αSyn + rAAV-AS69 compared to animals transduced with rAAV-αSyn + rAAV-GFP (Figure 3C), albeit not reaching statistical significance. The average count of P-αSyn-positive cells within the SNc was significantly higher in rAAV-αSyn + rAAV-AS69-expressing mice than in rAAV-αSyn + rAAV-GFP-expressing mice (Figure 3C). Since rAAV-AS69 could influence the aggregation dynamics of P-αSyn, we decided to analyze the correlation of H-αSyn to P-αSyn for each group separately. The counts for H-αSyn-positive cells and P-αSyn-positive cells were highly correlated in mice transduced with rAAV-αSyn + rAAV-AS69 (Figure 3E). Mice transduced with rAAV-αSyn + rAAV-GFP did not show a significant correlation (Figure 3D).

### 3.4. Microglia Are Activated in Mice Transduced with rAAV-αSyn + rAAV-AS69

Microglia react to αSyn pathology and change morphology upon activation. Residential microglia exhibit long filigree dendrites (Figure 4A, green arrow). Hypertrophic and dystrophic microglia (Figure 4A, yellow arrow) are considered as an activated subtype [26]. Amoeboid and rod-shaped microglia are also associated with neurodegenerative processes and can be recognized by an enlarged soma with ruffled extensions (Figure 4A, cluster of microglia: red arrow). In order to investigate microglia activation in our model, we quantified and categorized Iba1-positive cells in the SNc. 

In mice transduced with rAAV-GFP, rAAV-αSyn + rAAV-GFP or rAAV-αSyn + rAAV-AS69, microglia numbers were 50% higher in the injected hemisphere as compared to the uninjected hemisphere (Figure 4B), suggesting that rAAV transduction caused microglia activation. With respect to microglia subtypes, about 75% of Iba1-positive microglia in mice transduced with rAAV-GFP, rAAV-GFP + rAAV-AS69 or rAAV-αSyn + rAAV-GFP were in a residential state (Figure 4C). This fraction was lower in mice transduced with rAAV-αSyn + rAAV-AS69. Conversely, activated microglia subtypes were significantly more common in mice transduced with rAAV-αSyn + rAAV-AS69 than in the other groups (Figure 4C). The counts for hypertrophic microglia correlated with the count of P-αSyn-positive cells (Figure 4D), consistent with the hypothesis that both effects might be causally linked.

#### Astrogliosis Is Not Attenuated by rAAV-AS69

In order to quantify astrogliosis, we determined the GFAP-positive area in the SNc (Figure 4A). In mice transduced with rAAV-GFP or rAAV-GFP + rAAV-AS69, the size of the GFAP-positive area in the injected hemispheres was similar to the uninjected hemisphere (Figure 4E). In mice transduced with rAAV-αSyn + rAAV-GFP or rAAV-αSyn + rAAV-AS69, the GFAP-positive area was 15% larger in the injected hemisphere than in the uninjected hemisphere (Figure 4E), suggesting that αSyn expression could contribute to astroglia activation whereas AS69 transduction had no effect.

## 4. Discussion

In this study, transduction of SN neurons with rAAV-AS69 did not reduce the extent of neurodegeneration resulting from the overexpression of αSyn.

### 4.1. Degeneration of the Dopaminergic Neurons

The transduction of SN neurons with rAAV-αSyn caused αSyn pathology as evidenced by cells positive for H-αSyn and P-αSyn (Figure 3C), degeneration of dopaminergic neurons (Figure 2B,C) and a motor phenotype (Figure 1B). This finding is consistent with previous studies using the same paradigm [22]. Even though aSyn transduction affected mouse behavior in the cylinder test (Figure 1B), we did not find statistically significant degeneration of TH-positive axon terminals in the striatum (Figure 2E). We attribute the discrepancy to the modest degree of degeneration of TH-positive neurons in the SNc. A reduction of 20% in the count of dopaminergic neurons, albeit significant, is somewhat modest compared to previous experiments [22]. We and others observed a decrease of up to 80% of TH-positive neurons in the SN in the past [22,27]. Furthermore, due to the viral injection, we observed non-homogenous degeneration in the SNc.

### 4.2. Microglial Response to Transduction with rAAV-αSyn and rAAV-AS69

The activation of microglia was most pronounced in mice transduced with rAAV-αSyn+ rAAV-AS69 (Figure 4A). Because the constitutive activation of inflammatory changes is sufficient to induce αSyn pathology [28], this finding would be consistent with the hypothesis that any reduction in αSyn pathology resulting from AS69 co-expression was consumed by the consequences of increased neuroinflammation. rAAV-AS69 alone did not activate microglia (Figure 4C), suggesting that the microglial activation observed with rAAV-αSyn and rAAV-AS69 could result from an altered processing of αSyn with AS69. Indeed, neuronal expression of AS69 did rescue the behavioral phenotype of neuronal αSyn expression in drosophila [4], which lack neuroinflammation by microglia. Interestingly, the increased microglial response in mice transduced with rAAV-αSyn+ rAAV-AS69 could have accelerated the disease and/or is a consequence of a higher P-αSyn burden (Figure 4D). It has been shown that microglia contribute to the propagation of αSyn, whereas a microglial inhibition proved to be protective [29]. Furthermore, amoeboid and rod-shaped microglial cells are associated with phagocytic activity and antigen representation [15]. Their relative increase in mice transduced with rAAV-αSyn+ rAAV-AS69 (Figure 4C) would again argue in favor of differences in αSyn processing due to rAAV-AS69.

### 4.3. Transduction with rAAV-AS69 Did Not Reduce αSyn Pathology

The transduction with rAAV-AS69 did not reduce αSyn pathology as quantified by H-αSyn and P-αSyn staining (Figure 3C); rAAV-AS69 also did not reduce the degeneration of dopaminergic neurons (Figure 2B,C). These findings were unexpected given the reduction in αSyn pathology we observed with AS69 in cultured neurons and mouse striatum [5]. This discrepancy could be due to a variety of factors. In our previous work, αSyn pathology was induced by intracerebral injection of PFFs, whereas αSyn overexpression was used in the present study. These models recapitulate different mechanisms of amyloid formation. The PFF-based models are mostly reliant on secondary nucleation. Through the injection of in vitro derived aggregates which act as “seeds”, one induces the misfolding of endogenous αSyn [30,31]. Even the kinds of PFF or “strain” (e.g., oligomers and fibrils) display different patterns of propagation throughout the brain and differ in the structure of the induced aggregates [32]. The rAAV-A53T-αSyn model is based on overexpression of monomeric αSyn which is increased up to twentyfold. Even in cases of familial PD due to *SNCA* locus duplication or triplications, mRNA levels were only increased twofold [9,33,34]. Indeed, AS69′s effect is dependent on its concentration. In order to effectively inhibit fibril elongation, it requires a 1:1 ratio of AS69 to monomeric αSyn, whereas the inhibition of secondary nucleation and lipid-induced nucleation is also effective at substoichiometric ratios [4]. Differences in transgene expression potentially favored αSyn expression instead of AS69, which potentially overwhelmed rAAV-AS69 capabilities. On the other hand, it has been shown that AS69 also inhibits aggregation at substoichiometric concentrations [4].

Collectively, these findings would be consistent with the hypothesis that PFF-based models do not capture the entire spectrum of αSyn pathology and toxicity. Along similar lines, we and others previously found that inhibiting αSyn fibril formation by synthetic mutations did not block αSyn toxicity [35,36,37].

### 4.4. Potential Increase in Synuclein Pathology Due to rAAV-AS69

We observed more P-αSyn-positive cells in the SNc of animals transduced with rAAV-αSyn + rAAV-AS69 compared to rAAV-αSyn + rAAV-eGFP. Since the count of H-αSyn-positive cells was similar, one can assume equal transduction efficiency. The increase in P-αSyn-positive cells could therefore be a consequence of changes in αSyn processing due to the transduction with rAAV-AS69. One could hypothesize that those aggregates that did form—even in the presence of rAAV-AS69—did not reach their usual size since AS69 eventually interfered. As a result, low-molecular-weight aggregates such as oligomeric aggregates could become more abundant. Indeed, in vitro experiments showed that aggregates that develop even in the presence of amyloid aggregation inhibitors are soluble and of low molecular weight [38]. On the other hand, the number of H-αSyn-positive cells almost reached statistical significance, which is why we cannot rule out viral vector-related effects. Furthermore, it requires further experiments to characterize aggregates in mice transduced with rAAV-αSyn + rAAV-AS69.

### 4.5. Comparison of AS69 to Other αSyn Aggregation Inhibitors

αSyn pathology spreading to non-neuronal cells might be rescued insufficiently by the expression of rAAV-AS69 with the neuronal promoter we used. Administering AS69 protein showed beneficial effects in our previous experiment. Indeed, both small molecules, UCB0599 and Anle138b, that are currently tested in clinical trials have been administered systemically in various mouse models of synucleinopathies [39,40]. Oral Anle138b can prevent neurodegeneration and synuclein pathology in toxin-based models (rotenone and MPTP) as well as in transgenic mice that overexpress human A30P-α-Syn [41,42]. UCB0599 has been studied in transgenic mice that overexpress human wild type αSyn using intraperitoneal injections five times a week for 3 months [43]. Neither Anle138b nor UCB0599 have been tested in a model that is based on the viral overexpression of αSyn. 

### 4.6. AS69′s Structure and Stability

AS69 is a dimer of two identical subunits that are covalently linked by a disulfide bond between the Cys-28 residues of both subunits. Intracellular disulfide bonds can be reduced in a disulfide exchange reaction with excess glutathione [44]. Potentially, the dimeric complex can therefore not be formed. On the other hand, we previously observed protective effects of AS69 protein in PFF mice [5]. One advantage of AS69 protein over rAAV-AS69 is its presence in the extracellular space after intrastriatal injection. It was therefore presumably active in an additional environment and at the same time less exposed to redox reactions.

In summary, rAAV-AS69′s inhibitory activities did not reduce neurodegeneration in the SN resulting from αSyn overexpression, highlighting the importance of testing potential therapeutic strategies in a variety of animal models and studying multiple routes of drug administration.

## Figures and Tables

**Figure 1 biomolecules-14-00756-f001:**
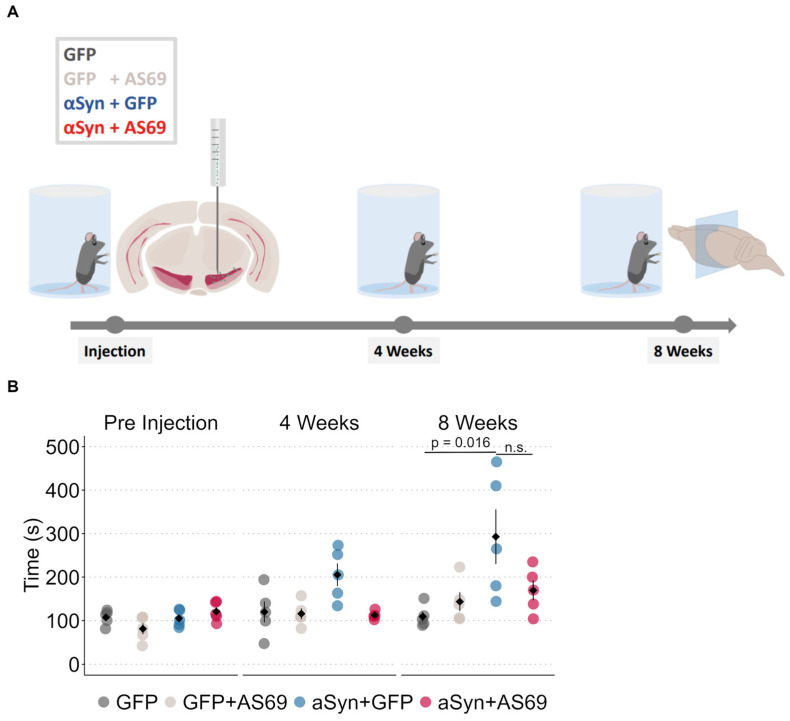
rAAV-AS69 does not attenuate αSyn-induced motor deficits. (**A**) Illustration of the study design. Five mice per group received unilateral stereotactic injections of rAAV into the substantia nigra pars compacta. Cylinder tests were performed one day before injection and four weeks and eight weeks after injection of rAAV. (**B**) Time required for 25 forepaw contacts with the glass cylinder wall at the indicated time points. Markers represent individual animals (*n* = 5). Kruskal–Wallis multiple comparisons followed by Dunn’s test as post hoc analysis, *p*-values were adjusted with the Holm method (*p*-value = 0.016 for GFP vs. αSyn + GFP). Non-significant results are not annotated or if shown are indicated by “ns”.

**Figure 2 biomolecules-14-00756-f002:**
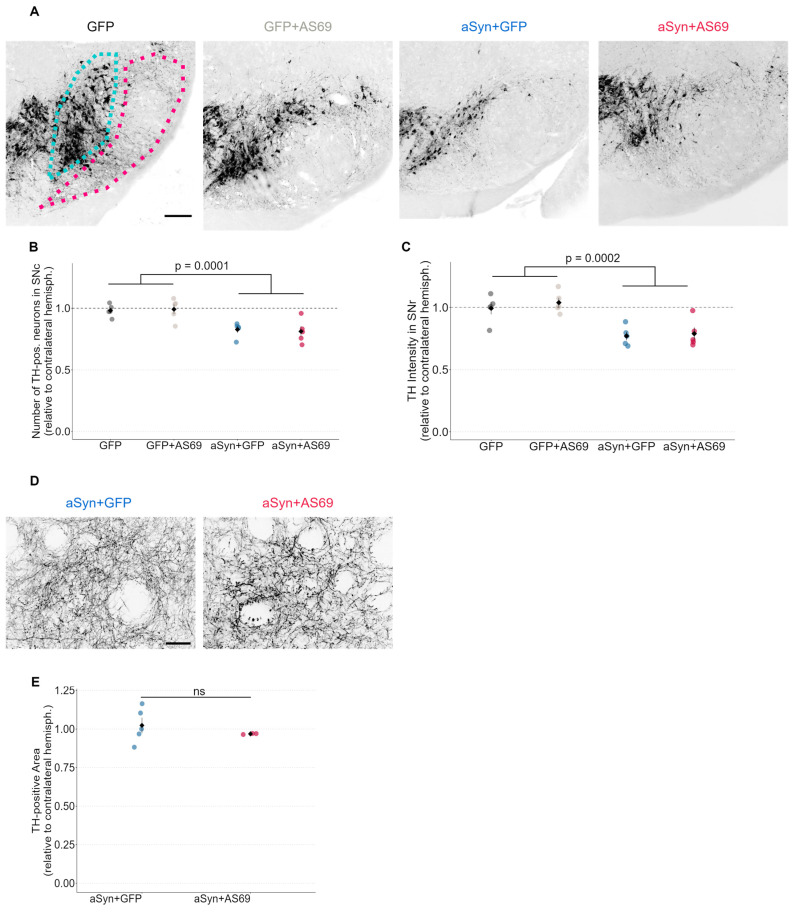
rAAV-AS69 does not prevent the degeneration of TH-positive neurons and dendrites. (**A**) Representative images for each cohort of the SN in the mouse midbrain stained for TH. Cohorts as indicated above each image. SNc is highlighted in blue and SNr is encircled in red. Scale bar: 200 μm. (**B**) Number of TH-positive neurons in the substantia nigra pars compacta relative to the contralateral hemisphere. Markers represent individual animals (*n* = 5). Comparison by two-way ANOVA (*p*-value = 0.0001 for factor GFP vs. αSyn; *p*-value = 0.942 for factor GFP vs. AS69). (**C**) Signal intensity of TH-positive dendrites in the SNr relative to the contralateral hemisphere. Labels represent individual animals (*n* = 5 animals per group). Comparison by two-way ANOVA (*p*-value = 0.0002 for GFP vs. αSyn; *p*-value = 0.531 for GFP vs. AS69). (**D**) Representative images of dopaminergic axon terminals, labeled by TH, in the striatum. Cohorts as indicated above each image. Scale bar: 30 μm. (**E**) TH-positive area in the striatum relative to the contralateral hemisphere. Markers represent individual animals (*n* = 5 for αSyn + GFP and *n* = 3 for αSyn + AS69). Non-significant results are annotated as “ns”.

**Figure 3 biomolecules-14-00756-f003:**
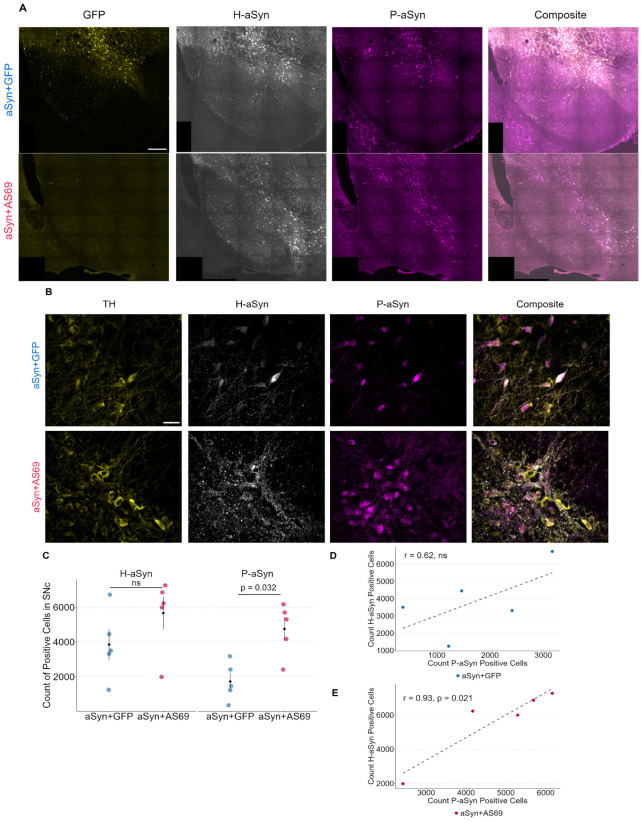
Transduction with rAAV-AS69 is associated with an increase in P-αSyn pathology. (**A**) Representative images of the SN showing GFP (yellow), H-αSyn (gray) and P-αSyn (magenta). Cohorts as indicated by row labels. Scale bar: 200 μm. (**B**) Representative high magnification images of TH-positive neurons (yellow), H-αSyn (gray) and P-αSyn (magenta) in the SNc. Scale bar: 30 μm. (**C**) Count of positive cells for H-αSyn and P-αSyn in the SNc of indicated cohorts. Markers represent individual animals (*n* = 5). Comparisons by Wilcoxon rank sum exact test (*p*-value = 0.22 for H-αSyn and annotated with non-significant “ns”; *p*-value = 0.032 for P-αSyn). (**D**) Correlation of H-αSyn and P-αSyn in animals transduced with rAAV-αSyn and rAAV-GFP. Pearson’s product–moment correlation at r = 0.63 (*p*-value = 0.26 and annotated with “ns”). Dotted line represents the corresponding generalized linear mixed model. (**E**) Correlation of H-αSyn and P-αSyn in animals transduced with rAAV-αSyn and rAAV-AS69. Pearson’s product–moment correlation at r = 0.93 (*p*-value = 0.021). Dotted line represents the corresponding generalized linear mixed model.

**Figure 4 biomolecules-14-00756-f004:**
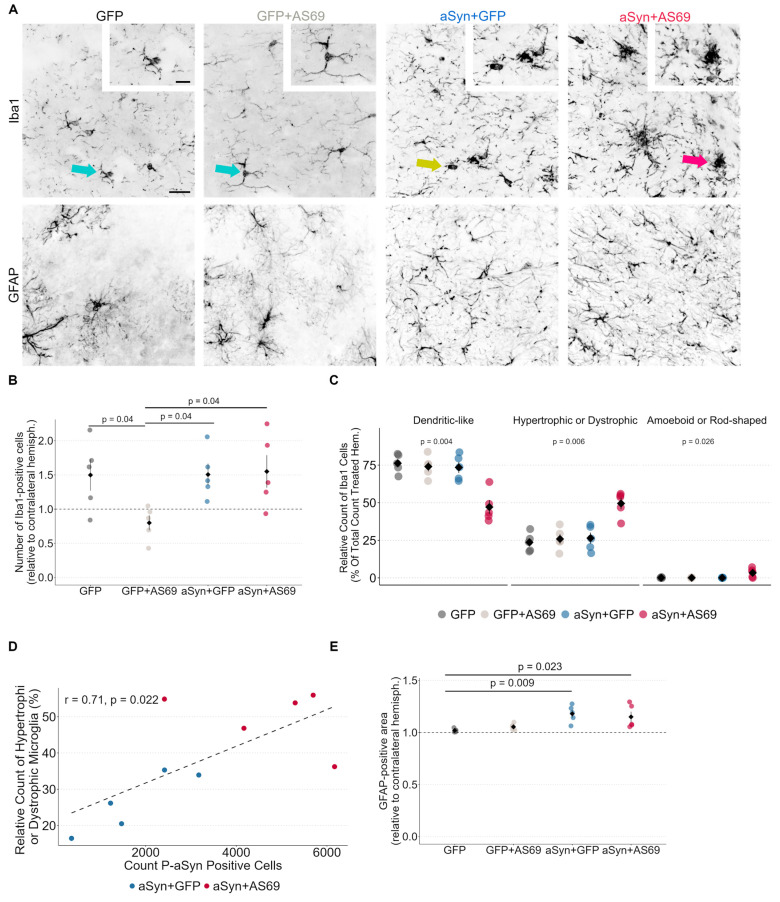
Microglia are particularly activated in αSyn + AS69-expressing mice and astrogliosis is not attenuated by rAAV-AS69. (**A**) Representative images of Iba1-positive microglia and GFAP-positive astrocytes in the SNc. Ramified microglia are highlighted by the green arrows. The yellow arrow indicates a dystrophic microglial cell and the red arrow shows an amoeboid microglial cell. Cohorts as indicated by the column names. Scale bar: 40 μm. Individual cells are shown in higher magnification in the inset. Scale bar: 20 μm. (**B**) Count of Iba1-positive cells in the SNc relative to the contralateral hemisphere. Labels represent individual animals (*n* = 5). Comparisons by multiple *t*-tests, *p*-values were adjusted for multiple testing with the Benjamini–Hochberg method (*p*-value= 0.04 for GFP vs. GFP + AS69, GFP + AS69 vs. αSyn + GFP and GFP + AS69 vs. αSyn + AS69). (**C**) Occurrence of each morphological microglia phenotype relative to the total count of microglia in the injected SNc. Labels represent individual animals (*n* = 5). Comparisons by two-way ANOVA followed by Tukey’s HSD (dendrite-like: *p*-value = 0.0008 for GFP vs. αSyn, *p*-value = 0.001 for GFP vs. AS69, *p*-value = 0.004 for the interaction; post hoc analysis: *p*-value = 0.0001 for GFP vs. αSyn + AS69, *p*-value = 0.0004 for GFP + AS69 vs. αSyn + AS69, *p*-value = 0.0005 for αSyn + GFP vs. αSyn + AS69; hypertrophic or dystrophic: *p*-value = 0.001 for GFP vs. αSyn, *p*-value = 0.001 for GFP vs. AS69, *p*-value = 0.007 for the interaction; post hoc analysis: *p*-value = 0.0003 for GFP vs. αSyn + AS69, *p*-value = 0.0007 for GFP + AS69 vs. αSyn + AS69, *p*-value = 0.0009 for αSyn + GFP vs. αSyn + AS69; amoeboid or rod-shaped: *p*-value = 0.029 for GFP vs. αSyn, *p*-value = 0.027 for GFP vs. AS69, *p*-value = 0.0256 for the interaction; post hoc analysis: *p*-value = 0.017 for GFP vs. αSyn + AS69, *p*-value = 0.016 for GFP + AS69 vs. αSyn + AS69, *p*-value = 0.015 for αSyn + GFP vs. αSyn + AS69). (**D**) Correlation of the count of P-αSyn-positive cells in the SN and microglia classified as “hypertrophic or dystrophic microglia” in the SN. Pearson’s product-moment correlation r = 0.71 (*p*-value = 0.022). Dotted line represents the corresponding generalized linear mixed model. (**E**) GFAP-positive area in the SNc relative to the contralateral hemisphere. Labels represent individual animals (*n* = 5). Kruskal–Wallis multiple comparisons followed by Dunn’s test as post hoc analysis, *p*-values were adjusted with the Holm method (*p*-value = 0.009 for GFP vs. αSyn + GFP, *p*-value = 0.023 for GFP vs. αSyn + AS69).

## Data Availability

The data presented in this study are available on request from the corresponding author.

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
