# Peer review of "Evaluation of the Effect of β-Wrapin AS69 in a Mouse Model Based on Alpha-Synuclein Overexpression"

_biomolecules, 2024, doi:10.3390/biom14070756_

Round 1

Reviewer 1 Report

Comments and Suggestions for Authors

Title: Evaluation of the effect of β-wrapin AS69 in a mouse model based on alpha-synuclein overexpression.

Summary

In this research article, Lennart Höfs et al. evaluated whether AS69 can reduce neurodegeneration resulting from a-synuclein overexpression. Human A53T-αSyn and AS69 were overexpressed in the mouse Substantia nigra (SN) by using recombinant adeno-associated viral vector (rAAV) transduction. Transduction with rAAV-GFP was used as a negative control. The results showed that transduction with rAAV-αSyn resulted in αSyn pathology, which caused degeneration of dopaminergic neurons in the SN, while the co-expression of rAAV-AS69 did not reduce αSyn pathology or degeneration of dopaminergic neurons. However, this manuscript is not publishable in Biomolecules.

Major comments are given below:

1. Select better keywords for this manuscript. The resolution of figures and images should be improved.

2. Figure 1B, for the cylinder tests at 8 weeks, is there a significant difference between αSyn + GFP vs. αSyn + AS69?

Statistical analysis results should be shown in the figures even though there is no significant difference.

3. -page 6, “The reduction of dopaminergic neurons by αSyn overexpression is therefore modest, but it can explain the subtle motor deficit (Fig. 1B).”

“Mice transduced with rAAV-αSyn + rAAV-GFP showed about 20% fewer TH-positive neurons than rAAV-GFP expressing animals.”

Does it mean that a 20% decrease is a modest change?

4. -Figures 2B and 2C, the figures should indicate which two groups show a significant difference.

5. -Figure 3A, why there are “grids” in the image background?

6. -Figure 3C, for the counting of positive cells in SNc, how many slices are used for the data collection?

Compared with the αSyn + GFP group, the p-Syn signal showed a significant increase in the αSyn + AS69 group. This indicates that the αSyn pathology may be worse when the AS69 is overexpressed. It is contrary to the previous findings that AS69 alleviates αSyn pathology. What are the possible explanations for this observation?

7. -Figure 3D, why the correlation between αSyn and p-Syn is important?

8. -for all the figures showing statistical analysis results, the data should be better presented. It is better to show the p-values in the figures, not the figure legends.

9. -Figure 4, how many slices from each animal are used for the counting? If only a small number of slices or images are used for analysis, the results may be biased.

10. Write more in the Discussion and this section should be better organized.

Comments on the Quality of English Language

Minor editing of English language required.

Author Response

Response to reviewer I

  1. Select better keywords for this manuscript. The resolution of figures and images should be improved.

We extended the list of keywords to improve searchability and visibility. We replaced the figures with high resolution svg-files.

  1. Figure 1B, for the cylinder tests at 8 weeks, is there a significant difference between αSyn + GFP vs. αSyn + AS69? Statistical analysis results should be shown in the figures even though there is no significant difference.

There is no significant difference between αSyn + GFP vs. αSyn + AS69 in figure 1B. As recommended, we added annotations in figure 1B, 2E and 3D indicating statistically non-significant differences.

  1. -page 6, “The reduction of dopaminergic neurons by αSyn overexpression is therefore modest, but it can explain the subtle motor deficit (Fig. 1B).” “Mice transduced with rAAV-αSyn + rAAV-GFP showed about 20% fewer TH-positive neurons than rAAV-GFP expressing animals.”

Does it mean that a 20% decrease is a modest change?

A loss of 20% of TH-positive neurons is a statistically significant change but in comparison to previous publications a modest. We and others previously reported a viral vector dose dependent loss of up to 80% of TH-positive neurons. The point discussed here could be interesting to other readers which is why we included it in the discussion.

  1. -Figures 2B and 2C, the figures should indicate which two groups show a significant difference.

As reported in the figure legend we performed a two-way ANOVA that showed a significant difference for one factor (transduction with either rAAV-GFP or rAAV-αSyn). The second factor (transduction with rAAV-AS69 or rAAV-GFP) was not statistically significant. Therefore, mice that were transduced with rAAV-αSyn showed a significant loss of TH-positive cells compared to mice transduced with rAAV-GFP (group GFP and GFP + AS69), which we annotated in Fig. 2B and 2C accordingly. Crucially, there was no significant difference in the post-hoc analysis between αSyn+GFP and αSyn+AS69.

  1. -Figure 3A, why there are “grids” in the image background?

For the analysis of H-aSyn and P-aSyn expression we imaged every fourth coronal brain section using a slide scanning microscope. These scans are based single images taken at 20x/0.8NA magnification which were then merged to one greater image/scan. The “grid” like appearance is due to the uneven illumination of each mosaic image. Crucially, as visible in the new high-resolution figures, individual cells and structures can be easily distinguished and were merged flawlessly across all channels.

6.1. -Figure 3C, for the counting of positive cells in SNc, how many slices are used for the data collection?

As described in the Method & Materials section we scanned about ten brain sections per animal. This means that every fourth brain section spanning the whole substantia nigra was used for this analysis.

6.2. Compared with the αSyn + GFP group, the p-Syn signal showed a significant increase in the αSyn + AS69 group. This indicates that the αSyn pathology may be worse when the AS69 is overexpressed. It is contrary to the previous findings that AS69 alleviates αSyn pathology. What are the possible explanations for this observation?

It could indeed mean that the co-expression of AS69 did worsen the pathology. We added a new paragraph in the discussion to address this question, since it could be highly interesting to other readers.

  1. -Figure 3D, why the correlation between αSyn and p-Syn is important?

Phosphorylated αSyn is a marker for pathological, aggregated αSynuclein. It is therefore important to show how the model behaves (Fig. 3D) and whether any changes occur due to the treatment with AS69 (Fig. 3E). Since AS69 could influence the aggregation dynamics of P-αSyn we decided to analyze the correlation for each group separately. We added this background information in the results section.

  1. -for all the figures showing statistical analysis results, the data should be better presented. It is better to show the p-values in the figures, not the figure legends.

As recommended, we replaced all asterisks by p-values in the figures.

  1. -Figure 4, how many slices from each animal are used for the counting? If only a small number of slices or images are used for analysis, the results may be biased.

As described in the Material & Methods section, we used 15-20 images across 4 brain sections per hemisphere of each animal, resulting in at least 30 images per animal. We therefore took 600 multichannel images (Iba1, GFAP). For image analysis channels were split to analyze GFAP and Iba1 separately which means that the results shown in figure 4 are based on the analysis of about 1200 images.

  1. Write more in the Discussion and this section should be better organized.

We restructured the discussion by introducing subheadings, inserting a new paragraph discussing the potential increase in pathology due to AS69 co-expression (see above 6.2.), by discussing more aspects in more detail and by referencing more up to date research/literature.

Reviewer 2 Report

Comments and Suggestions for Authors

The manuscript evaluated the potential neuroprotective effects of the protein β -wrapin AS69 in a mouse model based on alpha-synuclein overexpression. In particular, the authors assessed AS69 expressed by transduction with a recombinant adeno-associated vector (rAAV) - in a mouse model that is based on rAAV-mediated overexpression of human A53T-αSyn (H-αSyn) in the Substantia nigra (SN). The authors did not record neuroprotective effects in this in vivo experimental model. The co-expression of rAAV-AS69 did not reduce αSyn pathology or degeneration of dopaminergic neurons or astrogliosis.

The authors discussed only briefly the discrepancy of these findings compared to previous neuroprotective effects obtained with AS69 in a mouse model based on intrastriatal injection of aSyn pre-formed fibrils (Szegő et al. Front. Neurosci. 2021, 15), suggesting different redox reactions, at extracellular and intracellular, involved in the loss of efficacy of AS69. In this regard, since largely negative results, they should add new data explaining this discrepancy using in vitro or in vivo models. For example, an in vitro model of alpha-synuclein overexpressing mimicking alterations in the intracellular and extracellular redox state.

This information may be helpful to readers interested in biological drug development.

Author Response

Thank you for your suggestion. We appreciate your input and recognize the importance of understanding why rAAV-AS69 did not achieve the desired results. We went into great detail in the reworked discussion about why AS69 behaves differently in the PFF-model compared to our current results in the rAAV-model. However, we believe that conducting an experiment to investigate the specific flaws behind rAAV-AS69's performance falls outside the scope of this review. Our focus is on providing a comprehensive analysis of the current findings and on discussing their implications. We believe that the article is highly interesting to Biomolecule’s readership since it focuses on the structure, function, and development of AS69 which aligns perfectly with the journal's scope. Our work contributes valuable insights into the mechanisms of αSyn aggregation inhibition with implications for synucleinopathies and research.

Round 2

Reviewer 2 Report

Comments and Suggestions for Authors

The paper is suitable for pubblication